# Systematic Evaluation of Biotic and Abiotic Factors in Antifungal Microorganism Screening

**DOI:** 10.3390/microorganisms12071396

**Published:** 2024-07-10

**Authors:** Gunjan Gupta, Steve Labrie, Marie Filteau

**Affiliations:** 1Département des Sciences des Aliments, Université Laval, Quebec City, QC G1V 0A6, Canada; gunjan.gupta.1@ulaval.ca (G.G.); steve.labrie@fsaa.ulaval.ca (S.L.); 2Institut sur la Nutrition et les Aliments Fonctionnels (INAF), Quebec City, QC G1V 0A6, Canada; 3Institut de Biologie Intégrative et des Systèmes (IBIS), Université Laval, Quebec City, QC G1V 0A6, Canada

**Keywords:** microbial interactions, high-throughput methods, antifungal activity, *Pseudomonas*, yeast, bioprospection, microbial control applications, maple syrup industry

## Abstract

Microorganisms have significant potential to control fungal contamination in various foods. However, the identification of strains that exhibit robust antifungal activity poses challenges due to highly context-dependent responses. Therefore, to fully exploit the potential of isolates as antifungal agents, it is crucial to systematically evaluate them in a variety of biotic and abiotic contexts. Here, we present an adaptable and scalable method using a robotic platform to study the properties of 1022 isolates obtained from maple sap. We tested the antifungal activity of isolates alone or in pairs on M17 + lactose (LM17), plate count agar (PCA), and sucrose–allantoin (SALN) culture media against *Kluyveromyces lactis*, *Candida boidinii*, and *Saccharomyces cerevisiae*. Microorganisms exhibited less often antifungal activity on SALN and PCA than LM17, suggesting that the latter is a better screening medium. We also analyzed the results of ecological interactions between pairs. Isolates that showed consistent competitive behaviors were more likely to show antifungal activity than expected by chance. However, co-culture rarely improved antifungal activity. In fact, an interaction-mediated suppression of activity was more prevalent in our dataset. These findings highlight the importance of incorporating both biotic and abiotic factors into systematic screening designs for the bioprospection of microorganisms with environmentally robust antifungal activity.

## 1. Introduction

One third of the food produced for human consumption is either spoiled or wasted according to a report by the Food and Agriculture Organization of the United Nations [1]. Of this, about 20% are wasted due to fungal spoilage, which is equivalent to 260 million tons of food per year [2]. In view of growing consumer concern for food safety and environmental issues, the food industry is seeking biological alternatives to synthetic chemicals to inhibit undesirable microorganisms and thus extend the shelf life of food [3]. One such promising approach is the use of biocontrol agents or bioprotective cultures, which are naturally occurring microorganisms that can inhibit the growth of microorganisms in plants and foods, respectively [4]. These strategies have been shown to be effective in controlling fungal growth in food products such as fruits, vegetables, and dairy products [5].

The maple syrup industry is one of the examples in which microbial agents could be used to control fungal growth. Maple sap is commonly used to produce maple syrup, which is a highly valued product in the food industry [6]. It is collected during the spring season, when temperatures fluctuate above and below the freezing point [7], typically from thousands of maple trees (*Acer saccharum* Marsh.) through tubing networks. However, the networks are colonized by microbial biofilms due to the presence of sugars and other nutrients in the flowing sap, creating a nutrient-rich environment that supports the growth of bacteria, yeasts, and undesirable molds [8,9]. Traditionally, aseptic tapping techniques and chemical sanitizers are used to control fungal growth in these tubes [10]. However, the use of such agents raises concerns about potential chemical contamination and the introduction of off flavors in the final maple syrup product [8]. Additionally, maple syrup is subject to postproduction spoilage caused by contaminating yeast and molds that causes significant economic losses [11,12]. Therefore, the use of bioprotective agents or microbial antifungals endogenous to the maple environment could offer a promising and environmentally friendly alternative for managing fungal problems in the maple syrup industry. This approach could help maintain the quality of maple syrup, ensuring its marketability and reducing postproduction losses.

Maple sap presents a promising source for obtaining antifungal agents. The microbiota of this substrate has been well documented and is known to harbor a diverse range of microorganisms, including *Pseudomonas* spp., and less abundantly lactic acid bacteria (LAB) and psychrophilic yeasts [9,13]. Some of these microorganisms, such as *Pseudomonas* and LAB, are known for their production of various antimicrobial compounds, including cyclic lipopeptides and bacteriocins [14,15]. In particular, certain strains of *Pseudomonas* and LAB have a long history of safe use in the food industry and have been used as biocontrol and bioprotective cultures, respectively [5]. However, until now, the antifungal potential of microorganisms isolated from maple sap has remained untapped.

Microorganisms live in communities and therefore interact and engage in communication, competition, cooperation, and co-evolution with each other [16,17], which can result in suppression or induction of their antifungal effects [18,19]. Moreover, microorganisms are subject to abiotic perturbations, that is, changes in environmental conditions such as pH, temperature, nutrient availability, etc. [20,21]. This can have a significant impact on their ecological [17,22] and functional interactions [23,24]. In maple sap, for example, changes in nutrient content during early and late spring have been linked to changes in microbial association networks [9]. However, experimental evidence on the extent to which biotic and abiotic factors influence antifungal activity remains limited, and there are many conditions that are currently unexplored [5,21]. Therefore, a screening approach that integrates biotic and abiotic variations could help identify strains with desirable properties.

The identification of effective antifungal agents requires the presence of specific key characteristics. These microorganisms should demonstrate potent antifungal activity against a wide range of fungal contaminants and possess a competitive ability to colonize the target environment. Furthermore, their antifungal activity should remain stable even in the presence of other microorganisms in the environment. Additionally, they must be safe for use in food products, ensuring that their presence does not negatively impact the safety or sensory attributes. However, the search for such effective antifungal biocontrol agents has proven to be challenging, requiring extensive screening and evaluation processes [25]. High-throughput experiments have emerged as a powerful tool for rapidly screening a large number of microorganisms for antimicrobial activity [25]. In addition, automated culture-handling systems can help quickly assess the antifungal activity of thousands of microorganisms under various environmental conditions.

In this study, we used a high-throughput experimental approach using a robotic platform to assess the antifungal potential of maple sap isolates against common yeasts associated with food spoilage [26,27]. Our goal was to identify antifungal strains that are less influenced by biotic and abiotic changes. Thus, we studied the influence of co-culture and media composition on antifungal activity. By considering the contextual nature of antifungal activity, this research contributes to the development of targeted strategies for the identification of potential natural antifungal agents.

## 2. Materials and Methods

### 2.1. Maple Sap Isolates

Microorganisms were isolated from 15 maple sap samples collected during 2015 and 2016 from different regions in Canada that have been previously described in [9]. These samples were kept frozen at −80 °C until use in this study. Based on their previously characterized microbial composition, the following samples were selected: X2015.5, AJ1g, AJ2, D2, F2, F3, F4, GF3, HV2, HV5, JMN1, MM1, P2, rg2.17.b, and yb1.17b [9]. A 1:10 dilution of each sap sample was plated using sterile glass beads and incubated at 20 °C. To maximize diversity, three media were used for isolation i.e., M17 (Nutri-Bact, Terrebonne, QC, Canada) + 0.5% lactose (LM17), plate count agar (PCA, BD Difco, Thermo Fisher Scientific, Saint-Laurent, QC, Canada), and sucrose–allantoin (SALN) media (1.75 g/L of yeast nitrogen base without amino acids (BD, Thermo Fisher Scientific), 1.25 g/L of allantoin (Sigma-Aldrich, St. Louis, MO, USA), 2% sucrose). LM17 is a rich medium that favors the growth of lactic acid bacteria, while SALN is a minimal medium mimicking maple sap [28]. PCA is a general-purpose medium commonly used for bacterial enumeration that enables the growth of a wide range of microorganisms. M17 (Nutri-Bact) containing ascorbic acid, 5 g/L; magnesium sulfate, 0.25 g/L; meat extract, 5 g/L; meat peptone, 2.5 g/L; sodium glycerophosphate, 19 g/L; soya peptone, 5 g/L; tryptone, 2.5 g/L; and yeast extract, 2.5 g/L [29] was supplemented with 0.5% lactose. SALN contained 1.75 g/L of yeast nitrogen base, 1.25 g/L of allantoin, and 2% sucrose. PCA (Difco) contained enzymatic digest of casein/tryptone, 5.0 g/L; yeast extract, 2.5 g/L; glucose, 1.0 g/L; and agar, 15.0 g/L. Plates of LM17 and SALN media were prepared adding 2% agar. Nunc OmniTray plates (size 128 × 86 mm; cap. 90 mL; Nunc, Nalge Nunc International, Rochester, NY, USA Cat # 82-264728) were filled with 35 mL of media. Diluted sap samples were plated using sterile glass beads and incubated at 20 °C. Different maple sap samples were used for isolation in each medium. After 48 h of incubation at 20 °C, isolated colonies were picked using the BM3-BC robot build-in protocol (S&P Robotics Inc., North York, ON, Canada), organized in a 96-array format and further incubated at 20 °C for 72 h. Stock cultures of a total of 1022 isolates were prepared in LM17, SALN and PCA containing 20% glycerol (*V/V*), frozen and then stored at −80 °C.

### 2.2. PCR and 16S rRNA Gene Sequencing

A subset of microorganisms isolated on each culture medium was selected for identification. Colony PCR was performed to amplify the 16S rRNA gene. The colonies were slightly touched with a tip and resuspended in 15 μL sterile MQ water. The resuspended colonies were heated at 95 °C for 20 min. The tubes were centrifuged for 4 min at 15,000 rpm, and 2 μL of the supernatant of each bacterial isolate was transferred to a new PCR tube containing a PCR mix (10X PCR buffer, 25 mM MgCl_2_, 10 mM dNTP mix, 10 μM OligoF, 10 μM OligoR, 5 U/μL Taq DNA polymerase (Bioshop Canada, Burlington, ON Canada) [30]. The primers used for amplification were 27f (5′-AGA GTTT GAT CMT GGC TCAG-3′) and 1492r (5′-GRT ACC TTG TTA CGA CTT-3′). All PCR reactions were performed in a TOne Thermal Cycler (Biometra, Goettingen, Germany) with the following settings: initial cycle 94 °C for 5 min and 35 cycles of 94 °C for 30 s, 55 °C for 30 s and 72 °C for 1.5 min, and finally 3 min at 72 °C. The PCR products were sent to Plateforme de séquençage, CHU de Québec-Université Laval for purification and Sanger sequencing. Geneious version 2022.2.2 (Dotmatics, Boston, MA, USA) [31] was used to trim and align forward and reverse sequences to form the consensus sequence. Quality controls were carried out manually in a consensus sequence after visual inspection of the electropherograms. The sequences were then compared to the non-redundant nucleotide collection of the National Center for Biotechnology Information (NCBI) using the MegaBLAST algorithm [32] available on Geneious. For identification, hits with the highest bit score were selected.

### 2.3. Target Yeasts

The yeasts were grown on yeast extract peptone dextrose (YPD) consisting of 1% yeast extract, 2% tryptone, and 2% glucose. In this work, three target yeast strains were used, namely *Kluyveromyces lactis* (accession number KF057495) [33]; *Candida boidinii* (accession number KF057593) [33]; and *Saccharomyces cerevisiae* (*hoΔ*::*Kan*MX) from the collection of prototrophic yeast deletions of MATa [34]. This strain of *S. cerevisiae* is a model organism, and therefore, its use can facilitate the comparison of antifungal detection rates with other teams as well as facilitate further experiments to explore the mode of action of selected strains. We validated the growth of each yeast target in the medium used for antifungal screening (LM17, SALN, and PCA) to exclude possible inhibitory effects due to the culture medium.

### 2.4. High-Throughput Experimental Design and Spot-on-Lawn Antifungal Assay

A first round of high-throughput experiments was performed to detect isolates showing antifungal effects against *S. cerevisiae* on their respective isolation media, i.e., LM17, SALN, and PCA. For this experiment, 3198 random pairwise interactions were measured between 1022 isolates. First, all isolates were precultured at 20 °C for 2 days on the medium they were isolated from at a density of 96 spots per plate. Four different 96-array preculture plates were combined to make a 384-array plate to perform the antifungal assay (Figure 1a). For each media, we had one 384-array plate; these were named M-MSP (M17 Main Source Plate), S-MSP (SALN Main Source Plate), and P-MSP (PCA Main Source Plate). Another copy of the isolates was made in which we condensed one 96-array preculture plate into one 384-array plate containing four identical 96-quadrants having four replicates per isolate (Figure 1a). In this assembly of isolates, each media had four 384-format plates, which were named M-RSP1 to M-RSP4 for M17, S-RSP1 to S-RSP4 for SALN, and P-RSP1 to P-RSP4 for PCA where RSP stands for ‘Replicate Source Plate’. Isolates were then screened for their antifungal activity individually and in pairs. For the individual condition, a spot-on-lawn antifungal assay was performed twice with replicates on all 384-array plates. To generate the pairwise combinations, we replicated two 384-array plates on top of each other; therefore, each spot on the plate was inoculated with two isolates (Figure 1b). Specifically, M-MSP was paired with four M-RSPs, S-MSP was paired with four S-RSPs, and P-MSP was paired with four P-RSPs. For each medium, the four interaction plates contained the same sets of isolates, that is, community members, but in varying combinations and proportions. Therefore, these plates were considered as replicate microbial communities. In this experimental design, each isolate was grown in a pairwise combination with three other isolates and itself, and each pairwise combination was tested twice, however, these were on different plates, at different positions with different surrounding neighbors, and replicated in a different order. Due to these variations, we expected different interaction outcomes and, therefore, report each instance of pairwise combinations separately.

For the detection of antifungal effects, a high-throughput spot-on-lawn assay at a density of 384-spots per plate was performed. To make the lawn, the targeted yeast strains were cultured overnight in liquid YPD broth at 20 °C. After 24 h, the yeast suspensions were adjusted to an optical density at 600 nm (OD_600nm_) of 0.1 and then further cultured for 24 h until the final OD_600nm_ reached 0.5. Subsequently, a 2 mL volume of the 48 h yeast culture was spread uniformly in all directions on an OmniTray. Once the plates were completely dry, the test microorganisms were replicated on top and incubated at 20 °C (Figure 1c). As a growth control, the isolates were grown in parallel without the yeast lawn. All plates were imaged every two hours during the day for four days to measure the growth of isolates and the evolution of inhibition halos.

We then selected isolates for a second round of experiments based on specific criteria. These criteria included isolates that showed an antifungal effect against *S. cerevisiae*, had distinct morphologies, or displayed detectable ecological interactions. For the second round of experiments, 95 isolates from all three media that displayed antifungal activity against *S. cerevisiae* individually or in pairs were placed on a 384-array plate named AfSc (antifungal against *S. cerevisiae*) consisting of four identical 96 quadrants. Additionally, 62 isolates from LM17, 79 from PCA, and 83 from SALN were selected and condensed together with the AfSc plate in a 384-array plate called MSP-2. Some isolates were randomly repeated on the plate to fill in positions and keep the density similar between plates, but these replicates were not considered for further analysis. We screened MSP-2 isolates for their antifungal activity individually and in pairs, where the AfSc plate was replicated on top of the MSP-2 plate for the paired condition. This second round of experiments involved the selection of 262 pairwise interactions against *K. lactis*, *C. boidinii*, and *S. cerevisiae* on both LM17 and SALN. Finally, we confirmed the antifungal effect of selected isolates and pairs of both rounds of experiments using a similar assay in a 96-array plate format.

### 2.5. Image Analysis and High-Throughput Data Processing

While antifungal plates were manually analyzed using visible zones of inhibition, time-series images of control plates without any fungal lawn were analyzed using IRIS v0.9.7.1, which is an open-source image analysis software [35]. We used the default parameters of the “morphology&color” profile with the following modifications: “FixedCropping_X_Start”:470, “FixedCropping_Y_Start”:330, “FixedCropping_X_End”:4140, “FixedCropping_Y_End”:2750, and finally, “ColonyBreathingSpace” was set to 100. Although ‘colonies’, that is, spots were automatically detected in IRIS, we manually checked each image to ensure that only spots were detected and that edge traces were accurate.

Statistical and multivariate analysis was performed in JMP PRO software version 15.2 (SAS Institute, Cary, NC, USA). We started by calculating an estimate of biomass for each spot by multiplying the size and opacity parameters. We then applied a log_2_ transformation to the biomass. We modeled the growth curves by fitting a three-parameter Gompertz model using time-series data containing up to 20 data points. Since some spots were not consistently detected by the image analysis software, spots with fewer than five data points were excluded from the subsequent analysis. We retained only the models for which all three parameters of the growth curve (asymptote, growth rate, and inflection point) had a Chi-square *p* < 0.05. To ensure accuracy, we visually inspected the growth curves and removed any aberrant values. Following this data quality check, from the first round of experiments, we obtained 4976 and 3425 growth curves under individual and paired conditions, respectively. In the second round of experiments, we obtained 559 individual growth curves and 569 paired growth curves on both LM17 and SALN media. The parameters of the growth curve were then used to calculate the integrated fitness (IF) by integrating the area under the curve for up to 72 h of growth [36]. We normalized the dataset by centering the IF values to the mean of the plate using the robust centered function in JMP to account for plate-to-plate variation. To obtain a conservative estimate of variance between replicate spots, a standard deviation was obtained for each isolate from the IF values of four replicates. We then calculated the median standard deviation (MSD) of the entire dataset, excluding the spots at the border positions on the agar plates. We compared the IF value of a pair of microorganisms with the individual value obtained from the same position on separate plates. One value came from the RSP plate, while the other value came from the MSP plate. Despite the slight differences in neighboring microorganisms on these plates, this approach allowed us to account for any positional bias when comparing the paired condition to the corresponding individual value at the same position on their respective plates. Using these IF values, we conservatively classified pairwise interactions into three ecological categories: cooperation, competition, and neutralism (Figure 2). If the IF of the pair was greater than both individual IF plus twice the MSD, then the effect was positive, that is, “cooperation”. On the contrary, if the IF of the pair was smaller than the individual IF of both minus twice the MSD, then the effect was negative, that is, “competition”. Values falling in between were deemed “neutral”.

To assess functional interactions, we qualitatively scored visible zones of inhibition in a spot-on-lawn assay to measure antifungal effects in individual and paired conditions (Figure 2). If the pair exhibited an antifungal effect and neither individual isolate exhibited it, we classified the interaction as “induction”. Conversely, if one of the isolates displayed an effect, while the pair did not, we classified the interaction as “suppression”. Finally, if both individuals and the pair showed an effect or if neither did, we classified the interaction as “neutral”, since there was no functional difference.

## 3. Results

### 3.1. Impact of Abiotic Conditions on Antifungal Effects

A total of 1022 isolates were analyzed on their respective isolation media for their antifungal activity against *S. cerevisiae*. Of these, 49 (4.79%) isolates showed antifungal activity. The frequency of antifungal activity detected was not equally distributed on the different media (Log likelihood, Chi-square Prob < 0.0001); the rates were ranked in accordance with the richness of nutrients. Specifically, LM17 exhibited the highest proportion (n = 40, 11.14%), which was followed by PCA (n = 7, 1.86%) and SALN (n = 2, 0.67%). The antifungal activity of all LM17 and SALN isolates was later confirmed in a separate experiment. In a subsequent set of experiments, a subset of isolates (146 from LM17, 86 from PCA, and 87 from SALN) were tested for antifungal activity against *K. lactis*, *C. boidinii*, and *S. cerevisiae* on LM17 and SALN media. The Venn diagram in Figure 3a illustrates the number of antifungal isolates detected for each target by media. To estimate positional bias, we evaluated the individual isolate antifungal activity by randomizing them. We observed a 98% congruence between both replicates. We anticipated a higher prevalence of antifungal activity against *S. cerevisiae* than the other targets due to enrichment from the first round of experiments. However, we observed that 26 isolates exhibited an antifungal effect against all three yeast targets on LM17 medium. Notably, the likelihood of encountering isolates displaying activity against all three yeast targets was not equivalent across isolation media (Log likelihood, Chi-square Prob < 0.0001), with LM17 being higher than SALN or PCA. In fact, of these 26 isolates, 21 were initially isolated on LM17 medium (Appendix A). Therefore, not only was the frequency of antifungal activity higher on LM17, but isolation on this media selected for isolates antifungal against multiple targets. In contrast, only one isolate demonstrated antifungal activity against multiple targets on SALN medium, while 42 isolates exhibited activity against *S. cerevisiae*. Interestingly, 36 out of these 42 isolates were initially isolated on LM17 (Appendix A). Again, the likelihood of detecting anti-*S. cerevisiae* activity on both LM17 and SALN seems dependent on the composition of the isolation medium (Log likelihood, Chi-square Prob < 0.0001) with microorganisms selected on LM17 having a higher frequency. However, activity against *K. lactis* and *C. boidinii* was less prominent on the SALN medium, and the probability that isolates exhibited antifungal activity against these targets was not associated with their isolation media (Log likelihood test, Chi-square Prob > 0.0001) (Figure 3b). In general, these results suggest that antifungal activity against *S. cerevisiae* is less influenced by the abiotic conditions tested compared to activity against the other two yeast targets.

### 3.2. Impact of Biotic Conditions on Antifungal Effect

To test whether contact-dependent interactions would induce additional antifungal effects, we established 2463 random pairwise interactions between 1022 isolates in the first round of experiments. In total, 122 pairs (5.0%) were inhibitory against *S. cerevisiae*. Like for individual assays, antifungal activity rates of paired isolates were significantly different across media (Log likelihood test, Chi-square Prob < 0.0001). We observed that 9.9% of the isolates were antifungal on LM17 (n = 98), 2.3% on PCA (n = 22) and 0.4% on SALN (n = 2). Furthermore, by comparing antifungal activity in individual and paired conditions, we classified the outcome, that is functional interactions, into three categories: induction, suppression, and neutral. In the first round of experiments, most of the functional interactions, i.e., 2310 (93.8%) were neutral, 146 (5.9%) interactions suppressed the antifungal activity seen in individual isolates, and only 7 (0.3%) interactions induced the antifungal activity in pairs. To address how these effects were distributed across different abiotic contexts, we analyzed the results by culture medium (Figure 4). No cases of induction were observed in SALN. Two (0.2%) and five (0.5%) cases were observed in PCA and LM17, respectively. In LM17, PCA, and SALN, 122 (12.4%), 15 (1.6%), and 9 (1.7%) suppressive interactions were observed, respectively. This pattern of functional interactions was consistent in the second round of experiments where the frequency of interaction-mediated antifungal effect induction was significantly lower than that of interaction-mediated suppression (Log likelihood test, Chi-square Prob < 0.0001); however, the effect varied by yeast target (Appendix A). Therefore, while it is possible that microbial interactions can induce inhibitory activity, changes in biotic conditions were more likely to negatively impact antifungal activity in our dataset.

### 3.3. Patterns of Local Ecological Interactions

Interactions between microorganisms play a crucial role in shaping their functions and behaviors, such as their antifungal activity. Therefore, by considering the ecological context of microorganisms, a more comprehensive understanding of their biocontrol activity can be attained. We evaluated local, that is, contact-dependent ecological interactions between isolates by comparing an integrated fitness score for individual isolates and pairs. In the first set of experiments, most of the interactions were neutral (n = 1795, 72.9%), while 374 (15.2%) interactions were classified as cooperative and 294 (11.9%) as competitive. There was no statistical difference between the percentage of cooperative and competitive interactions (Chi-square test, *p* > 0.1). Further analyzing the results by medium, no statistical differences were found between both types of interaction (Chi-square test, *p* > 0.1) (Figure 5).

In the second set of experiments, we observed 39 (23.5%) competitive and 20 (12.0%) cooperative interactions in SALN, while LM17 had 30 (18.1%) competitive and 5 (3.0%) cooperative interactions. There was no statistical difference between the proportions of cooperative and competitive interactions in LM17 and SALN (Chi-square test, *p* > 0.1). However, there was more cooperation than expected by chance in SALN compared to M17 (Fischer exact test, *p* = 0.0375), while there was no significant difference between both media for competitive interactions (Fischer exact test, *p* > 0.1).

### 3.4. Effect of the Abiotic Context on Ecological and Functional Microbial Interactions

To characterize the relative stability of ecological and functional interaction outcomes under different abiotic conditions, during the second set of experiments, we compared 166 interactions in LM17 and SALN media (Figure 6a). Nearly half of the ecological outcomes were scored the same across media (51.8%). The change in competitive and cooperative interactions occurred more often than expected by chance (Fischer exact test, *p* < 0.0001); however, the difference was more pronounced for competitive (Fischer exact test, *p* = 3.5 × 10^−23^) as compared to cooperative (Fischer exact test, *p* = 1.35 × 10^−7^) interactions. In fact, the changes from neutral outcomes were three times more frequent for competitive interactions (34.4%) than for cooperative interactions (11.4%). Thus, in the conditions tested, the abiotic context affected relatively more competition than cooperation.

Differences in functional interaction outcomes were also observed between the two media for the three yeast targets (Figure 6b). However, most functional outcomes did not differ between media regardless of the target (*K. lactis* 70.5%, *C. boidinii* 79.3%, and *S. cerevisiae* 83.7%). The change in suppressive interactions occurred more than expected by chance in all yeast targets (Fischer exact test, *p* < 0.0001). Indeed, for *K. lactis* and *C. boidinii*, most interactions that were suppressive on LM17 were neutral on SALN (28.9% and 29.5%, respectively). However, for *S. cerevisiae*, only a fraction of the suppressive interactions differed in either direction, while 28.9% of the suppressive outcomes were consistent across media. Induction cases were almost never observed on both media, and the change was not significant (Fischer exact test, *p* > 0.1). Thus, these findings highlight the substantial impact of abiotic factors on the dynamics of ecological and functional interactions under the conditions tested.

### 3.5. Suppression of Antifungal Activity and Ecological Competition

We hypothesized that the observed suppression of antifungal activity could be attributed to ecological competition between isolates. In the first round of experiments, we observed no statistical difference in the probability of observing suppression among isolates engaged in competition, cooperation, or neutral ecological interactions (Fisher’s exact test, right-sided *p*-value > 0.1). In the second round of experiments, we obtained similar results except for the isolates that showed antifungal activity against *C. boidinii* on LM17 media. In this context, the probability of observing suppression was significantly higher in pairs that exhibited competition (Fisher’s exact test, right-sided *p*-value = 0.0272). Of a total of 55 pairs showing suppression, 15 pairs were classified under competition. Overall, our results indicate that only part of the observed suppression of antifungal activity could be associated with competition among the set of isolates tested. This suggests that other mechanisms of suppression may also be at play.

### 3.6. Antifungal Producers Tend to Be Competitive Isolates

In the first round of experiments, we investigated the ecological behavior of isolates by examining their interactions with three different partners. Specifically, our objective was to determine whether isolates that exhibited antifungal activity were more likely to engage in cooperation or competition. When analyzing the number of each type of interaction between antifungal and non-antifungal isolates, we observed a difference in the rates of competitive interactions (Log likelihood test, Chi-square Prob < 0.0001). Indeed, isolates that exhibited antifungal activity displayed a higher rate of competitive behavior in all three interactions compared to what would be expected by chance (Figure 7). When analyzed by media, this effect was significant in M17 (Log likelihood test, Chi-square Prob = 0.0002), but not in PCA or SALN (Log likelihood test, Chi-square Prob > 0.05). Furthermore, we did not observe a significant relationship between the antifungal activity and cooperative ecological behavior of isolates (Log likelihood test, Chi-square Prob > 0.05). Additionally, none of the antifungal isolates demonstrated a mixed ecological behavior, indicating that none exhibited both cooperative and competitive ecological interactions. Overall, these findings demonstrate that microorganisms with antifungal activity are more likely to exhibit consistent competitive behavior.

In addition, we classified the isolates on the basis of their antifungal activity against three different yeast targets. If an isolate showed antifungal activity against both *K. lactis* and *S. cerevisiae*, we classified it as having an intrafamily effect, since these yeasts belong to the same phylogenetic family. On the other hand, if the isolate showed antifungal activity against all three targets (*K. lactis*, *S. cerevisiae* and *C. boidinii*) or against *C. boidinii* and either *K. lactis* or *S. cerevisiae*, we classified it as having an interfamily effect, as *C. boidinii* belongs to a different phylogenetic family. We did not observe significant differences in the rates of competitive and cooperative interactions between non-antifungal and antifungal isolates across different phylogenetic spectrums (Log likelihood test, Chi-square Prob > 0.05). The results were not significant for both the LM17 and SALN media.

### 3.7. Culturomic Approach

To characterize our population, a subset of 82 isolates was identified at the genus level using PCR amplification and DNA sequencing of the 16S rRNA gene. The phylogenetic affiliation of each sequence was attributed using BLAST results for the closest cultured strain, and a phylogenetic tree was constructed on aligned 16S rRNA gene sequences (Appendix A). Out of 82 isolates, 46 belonged to *Pseudomonas*, 25 to *Rahnella*, 2 to *Hafnia*, 1 to *Yersinia*, 5 to *Leuconostoc*, 1 to *Carnobacterium*, and 2 to *Gluconobacter*. Most of the isolates showing an antifungal effect were identified as *Pseudomonas* spp. In addition to *Pseudomonas*, only one (isolate G51), identified as *Leuconostoc*, showed an antifungal effect against *S. cerevisiae* on LM17.

Ultimately, we were interested in identifying strains with robust antifungal activity that can withstand different environmental conditions, exhibit competitive behavior, and maintain stable antifungal activity in pairwise assays (Appendix A). Among the isolates tested, only *Pseudomonas* sp. G78 consistently demonstrated robust antifungal activity against all three yeast targets on both LM17 and SALN media. However, 18 sequenced isolates showed antifungal effects against the three yeast targets on LM17 and *S. cerevisiae* on SALN. Interestingly, among these 18 isolates, 17 exhibited some degree of competitiveness against other isolates. Subsequently, we examined the suppression score of the isolates and made the surprising observation that the antifungal activity of all isolates was suppressed in at least one interaction. In fact, the antifungal activity of 77% of the sequenced isolates was suppressed during interactions (Appendix A).

## 4. Discussion

### 4.1. High-Throughput Assay Development

To meet the increasing demand for efficient high-throughput methods to screen the antifungal activity of microorganisms, we have developed a comprehensive workflow. This workflow incorporates an automated colony-picking and replicating system, allowing for the rapid screening of many microorganisms with enhanced precision and efficiency. Quantitative and qualitative image analysis techniques have been integrated to facilitate the analysis of macroscopic interactions, allowing the investigation of microbial interactions, identification of potential antifungal isolates, and the discovery of patterns of interest. The workflow can be adapted to different media, including natural matrices, target microorganisms, and growth conditions based on experimental requirements.

However, it is important to acknowledge some limitations associated with this approach. First, when studying multiple isolates or pairwise interactions on the same plate, potential higher-order effects from neighboring isolates or volatile compounds may arise [37]. Competition between different isolates can be influenced by local cell-to-cell interactions or greater species richness and diversity on the plate, thus impacting the outcome of local ecological interactions [21]. Second, inhibition halos were visually scored, which can be highly dependent on subjective evaluation. Lastly, we only considered qualitative changes, that is, the presence or absence of antifungal activity; therefore, it is likely that the reported suppression and induction rates are underestimates, since there could also have been quantitative changes. Future incorporation of software capable of quantitatively interpreting antifungal activity could improve workflow and result precision.

Despite these limitations, high-throughput microbial experiments, such as the one conducted here with multiple colonies per plate, provide a platform for investigating microbial functions or phenotypes as well as microbial interactions on a large scale in a community context. In fact, these plates can conceptually be considered as microbial ecosystems where microbial communities can be easily manipulated. Our pairwise design can also be augmented to increase the complexity of the biotic effect closer to that of a real microbiome. Thus, such high-throughput culture approaches provide new insights into how microbial communities function and can be adapted for various applications, including biocontrol, bioprotection, or biopreservation.

### 4.2. The Occurrence of Antifungal Effect Differs among Environmental Conditions

The selection of appropriate screening media is crucial in the study of microbial antifungal activity, as it significantly influences microbial communities [38]. The availability of nutrients, particularly sources of carbon and nitrogen, plays a central role in shaping the metabolic activities and physiological characteristics of isolates, thus affecting their antifungal potential [39,40]. In our study, we compared the antifungal activity of the same isolates on LM17 and SALN media and found that the isolates exhibited significant antifungal effects on LM17, while no detectable antifungal activity was observed on SALN against *K. lactis* and *C. boidinii*. However, the antifungal effect against *S. cerevisiae* was less affected by the abiotic conditions tested. Furthermore, most of the isolates showing antifungal activity were isolated on LM17 media. Therefore, the composition of LM17 appeared suitable both for the selection of antifungal isolates and for the expression of antifungal activity in maple sap isolates. However, it would be important to test the antifungal activity of microorganisms from other environments in LM17 to confirm if this observation can be generalized.

Variation in antifungal activity between LM17 and SALN could be attributed to different microbial populations present in different sap samples. Additionally, this difference could be explained by the specific presence or absence of nutrients in the culture medium. Indeed, LM17 is a nutrient-rich medium that is known to promote the growth of certain microorganisms, such as lactic acid bacteria (LAB). Previous studies have reported the suitability of LM17 as a screening medium for the antimicrobial activity of LAB. For example, Dodamani and Kaliwal reported that the M17 medium is best for the production of bacteriocin by *Lactococcus garvieae* [41]. Goh and Philip reported a high activity of bacteriocin produced by *W. confusa* in M17 supplemented with 1.5% glucose, sucrose, or lactose [42]. However, we predominantly isolated *Pseudomonas* sp. strains on LM17 medium from maple sap, and it was primarily these *Pseudomonas* sp. strains that exhibited antifungal effects. To our knowledge, the antifungal activity of *Pseudomonas* sp. strains in LM17 medium is a novel observation. In contrast, SALN is a minimal medium formulation that provides limited sources of nutrients required for microbial growth. The absence of protein and free amino acids in SALN hinders microbial growth, as microorganisms must rely on their own metabolism or that of other members of the community to obtain these essential amino acids. Moreover, the carbon source in SALN is sucrose, which requires the expression of specific enzymes such as invertase or sucrase for its utilization. The lack of these enzymes or their inhibition under certain conditions can limit microbial growth and consequently impact the production of antifungal compounds [43]. In addition, allantoin serves as the sole nitrogen source in SALN, and allantoin catabolism can cause metabolic changes that affect the expression of secondary metabolites. Navone et al. reported that the absence of negative feedback regulation on the allantoin pathway led to the accumulation and release of high levels of intracellular ammonium in *Streptomyces coelicolor*, which impaired antibiotic production [44]. These factors could contribute to the low antifungal activity observed in microorganisms grown on SALN medium. Furthermore, the low antifungal activity observed on SALN medium suggests a decrease in the antimicrobial potential of the microorganisms isolated in this study when grown in a culture media that mimics maple sap. Therefore, whether the antifungal activity is expressed under maple sap collecting conditions requires further testing. Overall, the contrasting antifungal effects observed between LM17 and SALN highlight the importance of understanding and optimizing nutrient conditions when studying microbial antifungal activity.

### 4.3. Competitive Nature of Antifungal Isolates

Ecological interactions between microorganisms play a fundamental role in ecosystem function [45], as extensively documented in the literature [46]. However, our understanding of these interactions in many environments remains limited. By integrating ecological knowledge into the strain selection process, we can significantly increase the likelihood of identifying and selecting suitable isolates for microbial control applications. In this study, we used a high-throughput pairwise interaction assay to investigate the ecological relationships among maple sap isolates. Our approach involved measuring the net output of interactions using conservative parameters. As a result, we were unable to capture the individual effects each partner had on the other, restricting our findings to interactions with only significant net positive or negative effects. While we did not observe significant differences between cooperative and competitive interactions, it was intriguing to observe that the antifungal isolates were more competitive than expected by chance. Antifungal isolates with competitive behavior were identified as *Pseudomonas* spp. The competitive nature of *Pseudomonas* has long been recognized as a key aspect of their biocontrol/bioprotective applications [47,48]. This competitiveness indicates their ability to effectively establish themselves and dominate the microbial community [49].

### 4.4. Impact of Interactions on Antifungal Behavior of Microorganisms

Drawing upon the understanding that microorganisms rarely live in isolation, we leveraged the pairwise interaction assay to examine the influence of ecological interactions on the antifungal activity of microorganisms. Our analysis revealed that ecological interactions predominantly resulted in the suppression of antifungal effects. This finding aligns with a high-throughput interaction study by Tyc et al. on soil bacteria, which reported a 22% occurrence of interaction-mediated suppression compared to a mere 6% occurrence of interaction-mediated induction [19]. Furthermore, additional research has shown a significant reduction in bacteriocin production by LAB in co-culture [18,50]. The high level of suppression during interactions may be attributed to various mechanisms, such as competition for essential resources and space, which limits the allocation of resources toward the production of antifungal compounds [51]. However, the result of our study only partially supports the view that the observed suppression of antifungal effects in co-culture can be attributed to direct competition between the two isolates. This suggests that other mechanisms may be at work. The regulation of antifungal production can be governed by quorum-sensing mechanisms [52]. In neutral interactions, we assume 50% growth of each partner in a co-culture, which may not be enough to reach the quorum needed to produce the antifungal effect. Previous studies have also shown that certain microorganisms can produce inhibitory metabolites or enzymes that directly interfere with antifungal activity. For example, proteases have been shown to degrade antifungal peptides, and enzymes can modify the structure of antifungal compounds, rendering them ineffective [53]. Additionally, the growth of microorganisms can induce changes in the physical environment, creating unfavorable conditions for the expression of antifungal activity. For example, pH modulation could modify the antifungal activity [5,54]. Another possible mechanism involves interference with signal transduction pathways needed for the production of antifungal compounds [55,56,57]. More research is required to elucidate the specific mechanisms underlying the observed suppression of antifungal effects in the interaction assay. However, these findings highlight the importance of understanding the complex interactions between microorganisms when developing microbial control strategies. These results also help identify the microorganisms that are incompatible together for microbial control applications. Indeed, our finding indicates that even if a potential biocontrol strain demonstrates robust activity in vitro, its antifungal activity can be suppressed during interactions with other microorganisms in situ. Therefore, depending on the specific biocontrol application, these patterns of antifungal activity and suppression can help to select the most suitable isolate. Lastly, the infrequent observation of interaction-mediated induction of the antifungal effect suggests that to harness the potential of coculture-induced antifungals, numerous combinations must be screened in high-throughput assays to identify interesting pairs for further investigation and potential applications.

### 4.5. Effect of the Abiotic Context on Ecological and Functional Microbial Interactions

Naturally, interactions between a pair of isolates are expected to vary significantly depending on the given growth environment [58,59]. The results of our study provide valuable information on the relative stability of the ecological and functional interaction outcomes under different abiotic conditions, i.e., in LM17 and SALN media. Approximately half of the ecological outcomes remained unchanged, indicating a level of stability, at least in terms of net positive or net negative interactions. Interestingly, only a small percentage of interactions remained cooperative or competitive across both media (0.6% and 2.4%, respectively), indicating a major shift in interaction types, or more likely strength, from cooperative or competitive to neutral, particularly from SALN to LM17. Indeed, a shift caused by a difference in interaction strength could mean a detectable net interaction in one condition but not in the other. These changes could be explained by the difference in media composition. Similar shifts in species interactions due to changes in nutrient availability have been reported [59,60]; however, these are on a small scale. Our results provide the relative proportion of interactions influenced by differences in abiotic conditions, contributing to a better understanding of the context dependence of these microbial interactions.

The selection of media also had an impact on the outcomes of functional interactions. Although most functional interactions remained stable across both media, a notable shift was observed from suppression to neutral when LM17 was compared with SALN for *K. lactis* and *C. boidinii*. This change could be attributed to the absence of antifungal effects on SALN for these two yeast targets. Overall, these findings provide intriguing patterns; however, the context dependence of microbial interactions underscores the need for further research to elucidate the underlying mechanisms that govern microbial ecology.

### 4.6. Maple Sap as an Ecological Niche

Our study demonstrates that the microbiome of maple sap has the potential to produce antifungal compounds that can be effective against a variety of yeast species. As expected from the previous metataxonomic analysis of the samples [9], *Pseudomonas* was the most isolated genus and played a significant role in the production of antifungal effects. *Pseudomonas* species are known for their ability to produce a variety of antifungal compounds, including phenazines, pyrrolnitrin, pyoluteorin, and cyclic lipopeptides such as pyoverdine and syringafactin [61]. Furthermore, some *Pseudomonas* produce antifungal compounds exhibiting certain degrees of thermal stability [62]. In the context of maple syrup production, where postproduction contamination is a concern, the application of heat-resistant antifungals holds substantial promise. This could offer a natural means to extend the shelf life and improve the safety of the syrup without the need for additional preservatives or chemical interventions. Interestingly, Filteau et al. have linked the presence of *Pseudomonas* with the quality of maple syrup [63]. Indeed, *Pseudomonas* spp. of the *fluorescens* complex showed a positive association with distinct attributes of maple syrup, such as maple and vanilla flavors. Therefore, by selecting desirable antifungal strains, understanding their interactions within the microbial community and the conditions that favor positive impacts on the quality of the syrup, it could be possible to use this knowledge for more controlled and optimized maple syrup production processes.

## 5. Conclusions and Outlook

Our study highlights maple sap as a rich reservoir of microorganisms, particularly *Pseudomonas* spp., with significant potential as biocontrol agents against fungal spoilage. Using a high-throughput assay, we systematically evaluated maple sap isolates for their antifungal activity and explored their ecological and functional interactions. This approach allowed us to investigate fundamental questions in microbial ecology, gaining valuable insights into members of the maple sap community. Furthermore, the identified antifungal isolates hold significant potential for microbial control applications. However, follow-up studies are essential to confirm activity in situ and to elucidate the mechanisms of action that underlie the antifungal activity of identified isolates. Additionally, it is essential to assess their antifungal activity against a broader fungal spectrum, especially targeting undesirable molds in maple sap tubing systems. Our experimental findings support the notion that the antifungal potential of these isolates is often influenced by both biotic and abiotic factors in their environment. However, it remains crucial to determine whether these trends extend across phylogenetic groups, other microbial ecosystems, and different nutrient environments. Overall, by shedding light on the context-dependency of antifungal potential, our work paves the way for future research aimed at harnessing the full potential of microorganisms for agriculture or other agrifood industries.

## Figures and Tables

**Figure 1 microorganisms-12-01396-f001:**
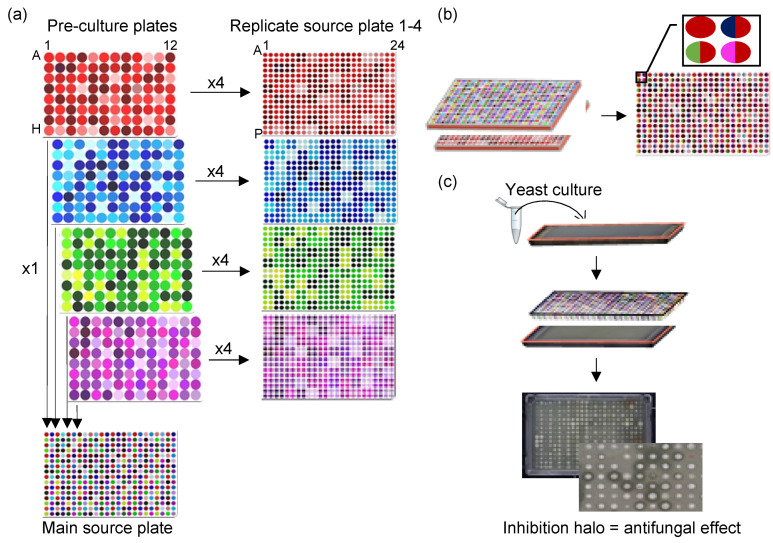
Workflow of high-throughput assays. (**a**) Overview of the assembly and design of the source plates. Using a 96-pin tool, each 96-array preculture plate was replicated four times into a 384-array plate, yielding four replicates per isolate (replicate source plate 1–4). In another assembly, the four 96-array preculture plates were merged to create a single 384-array plate (main source plate). The same assembly was performed separately for three culture media (PCA, LM17, SALN). (**b**) Overview of interaction plate assembly and design. To generate contact-dependent pairwise combinations, a replicate source plate and the main source plate were replicated on top of each other. Thus, each isolate was grown in pairs with three other isolates and once with itself. The same assembly was repeated for each culture medium. (**c**) Schematic representation of the high-throughput antifungal assay. A spot-on-lawn assay was performed to detect the antifungal effect of isolates or pairs. Specifically, a yeast culture was spread on agar media, and an array was replicated on top of the dried lawn. Antifungal activity was determined by observing visible zones of inhibition around the spots.

**Figure 2 microorganisms-12-01396-f002:**
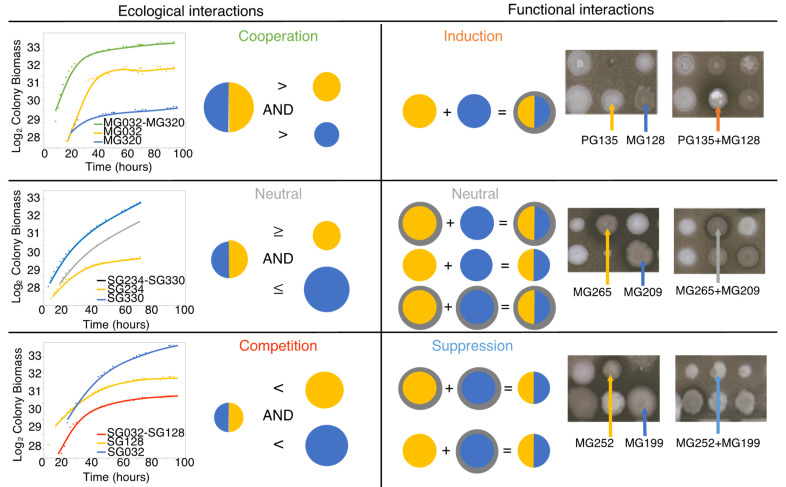
Summary of pairwise microbial interaction assays. For each interaction partner, there were three possible ecological and functional outcomes. Ecological interactions were scored using the integrated fitness, that is, the area under the modeled growth curve. The difference in integrated fitness between individual isolates and pairs was used to categorize ecological interactions. Antifungal activity was identified by visible inhibition halo around the colonies. The difference in antifungal activity between individual isolates and pairs was used to classify functional interactions.

**Figure 3 microorganisms-12-01396-f003:**
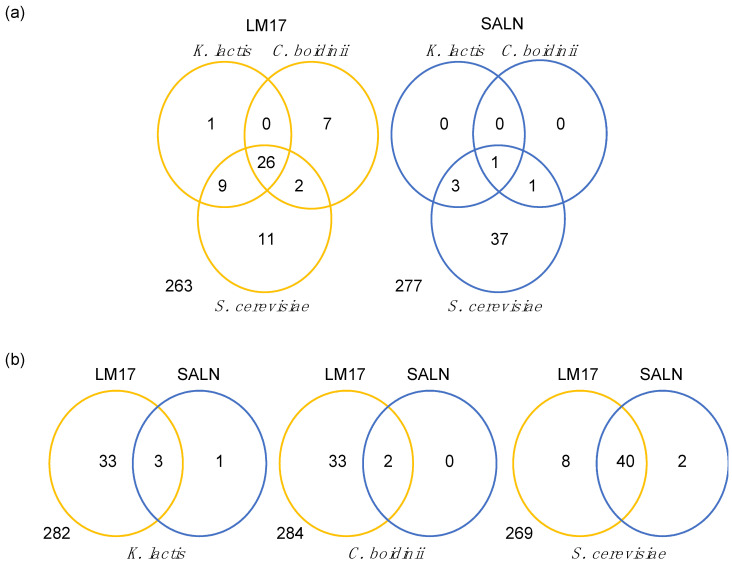
Venn diagram showing the number of isolates exhibiting antifungal effects (**a**) against the three target yeasts in each culture media and (**b**) in two culture media for each target yeast.

**Figure 4 microorganisms-12-01396-f004:**
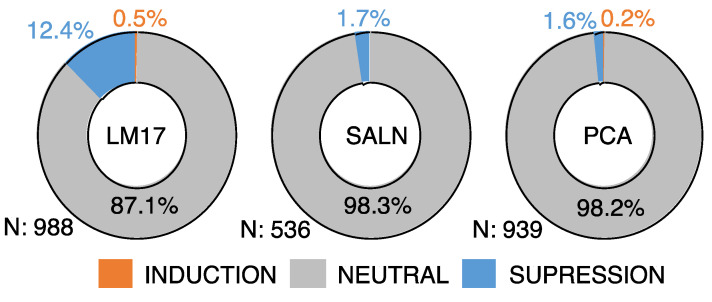
Proportions of interactions that induced or suppressed antifungal activity against *S. cerevisiae* in the first round of experiments. The frequency of interaction-mediated induction of the antifungal effect was significantly lower than interactions-mediated suppression (Log likelihood test, Chi-square Prob < 0.0001).

**Figure 5 microorganisms-12-01396-f005:**
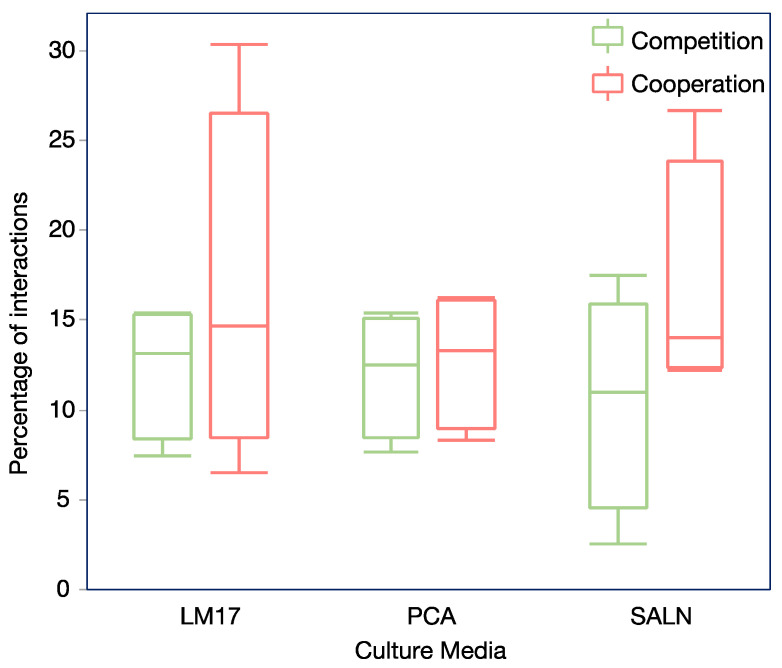
Box plot showing the percentage of cooperative and competitive interactions on each medium (N (LM17, PCA, SALN) = 4 plates). No statistical difference between the proportion of cooperative and competitive interactions was observed (Chi-square test, *p* > 0.1).

**Figure 6 microorganisms-12-01396-f006:**
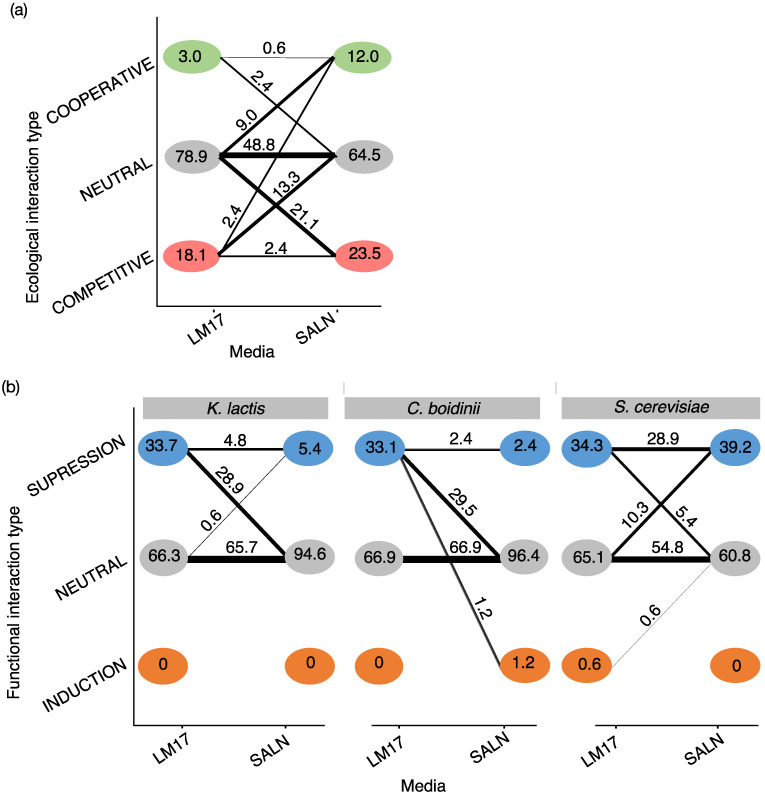
(**a**) Comparison of ecological interactions between LM17 and SALN. (**b**) Comparison of functional interactions between LM17 and SALN for each yeast target. For (**a**,**b**), each node represents an interaction outcome. The numbers in the nodes indicate the total percentage observed by media. The edge width and numbers represent the percentage of interactions that differ between LM17 and SALN.

**Figure 7 microorganisms-12-01396-f007:**
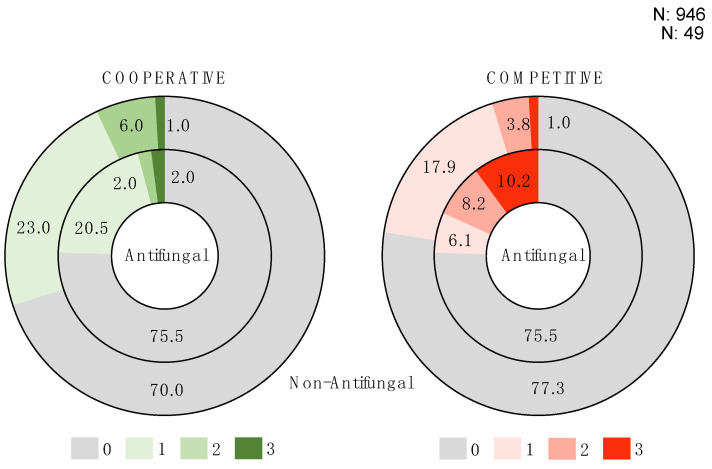
Pie charts illustrating the proportion of isolates exhibiting cooperative and competitive ecological behavior. The ecological behavior of each isolate was tested in three different interactions. The numbers (0, 1, 2, 3) represent the occurrences of cooperative and competitive interaction outcomes for each isolate. Non-antifungal isolate proportions are shown on the outer ring (N = 946) and antifungal isolate proportions are shown in the inner ring (N = 49). Antifungal isolates exhibited a higher frequency of competitive occurrence = 3 than expected by chance compared to non-antifungal isolates (Log likelihood test, Chi-square Prob < 0.0001).

## Data Availability

The images generated during this study are available on Figshare at DOI:10.6084/m9.figshare.26097436. The sequences obtained during this study were deposited in NCBI GenBank under accession number PP983064 to PP983147.

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
