# Peer review of "Systematic Evaluation of Biotic and Abiotic Factors in Antifungal Microorganism Screening"

_microorganisms, 2024, doi:10.3390/microorganisms12071396_

Round 1
Reviewer 1 Report
Comments and Suggestions for Authors
Reviewer 2 Report
Comments and Suggestions for Authors
I have revised the manuscript entitled “Systematic Evaluation of Biotic and Abiotic Factors in Antifungal Microorganism Screening”
This manuscript is very interesting, and has an approach to the process and problems in the production of maple syrup and the microorganisms found in it. At first, was difficult to provide a review because there are no number lines in the rows. In general the manuscript is well written and explained also possess a good information in detail for this area.
Introduction: Is well explained.
Methodology: In general, is very descriptive and understandable. Also refers modern methods of high quality and reliability. But it should be clarify:
Maple sap isolates:
The method of plate count or isolation is not clear. The sample was first diluted and then plated? Or why you say that the sample was frozen at -80? Please rephrase the method step by step and make it clear. If the culture media was purchased ready, it is not necessary to detail what it contains, only if you added an extra ingredient to the culture should be detailed.
It is not clear how they diluted the maple SAP samples. Dilutions 1:10? it is not clear. Please clarify in the text.
Results: The results are very descriptive and complex, are well explained contain details and illustrations. Are enough and pertinent.
Discussion: are very long and complex, but are well explained, are justify according to the results presented.
Conclussions: Are very good.
References: They need to be reviewed, there are not in the same font and there are not according to the format. They should be listed in appearance order and in the text should be referenced as a number. Please check and correct that part.
Reviewer 3 Report
Comments and Suggestions for Authors
The manuscript titled with " Systematic Evaluation of Biotic and Abiotic Factors in Antifungal Microorganism Screening ". The manuscript discusses a good point. Overall, the presented study indicates for used a high-throughput experimental approach using a robotic platform to evaluate the antifungal potential of maple sap isolates against common yeasts associated with food spoilage and identify antifungal strains that are less influenced by biotic and abiotic changes. Thus, the authors studied the influence of coculture and media composition on antifungal activity. By considering the contextual nature of antifungal activity, this research contributes to the development of targeted strategies for identifying potential natural antifungal agents. The manuscript written well. But it needs a minor revision in following:
-- MATERIALS AND METHODS
1- there are no sources or descriptions of the materials of media used for isolation
2- in method of PCR and 16S rRNA Gene Sequencing there are no sources or descriptions of the materials used for isolation and identification
- Minor editing of English language required
Comments on the Quality of English LanguageMinor editing of English language required
